# Breakthrough Technologies Reshape the Ewing Sarcoma Molecular Landscape

**DOI:** 10.3390/cells9040804

**Published:** 2020-03-26

**Authors:** Carmen Salguero-Aranda, Ana Teresa Amaral, Joaquín Olmedo-Pelayo, Juan Diaz-Martin, Enrique de Álava

**Affiliations:** 1Institute of Biomedicine of Sevilla (IBiS), Virgen del Rocio University Hospital/CSIC/University of Sevilla/CIBERONC, 41013 Seville, Spain; csalguero-ibis@us.es (C.S.-A.); amonteiro-ibis@us.es (A.T.A.); jjolmedo-ibis@us.es (J.O.-P.); jdiaz-ibis@us.es (J.D.-M.); 2Department of Normal and Pathological Cytology and Histology, School of Medicine, University of Seville, 41009 Seville, Spain

**Keywords:** sarcoma, Ewing sarcoma, GF, extracellular vesicles, circulating tumor DNA/RNA, DNA repair, breakthrough technologies, mutations

## Abstract

Ewing sarcoma is a highly aggressive round cell mesenchymal neoplasm, most often occurring in children and young adults. At the molecular level, it is characterized by the presence of recurrent chromosomal translocations. In the last years, next-generation technologies have contributed to a more accurate diagnosis and a refined classification. Moreover, the application of these novel technologies has highlighted the relevance of intertumoral and intratumoral molecular heterogeneity and secondary genetic alterations. Furthermore, they have shown evidence that genomic features can change as the tumor disseminates and are influenced by treatment as well. Similarly, next-generation technologies applied to liquid biopsies will significantly impact patient management by allowing the early detection of relapse and monitoring response to treatment. Finally, the use of these novel technologies has provided data of great value in order to discover new druggable pathways. Thus, this review provides concise updates on the latest progress of these breakthrough technologies, underscoring their importance in the generation of key knowledge, prognosis, and potential treatment of Ewing Sarcoma.

## 1. Introduction 

Ewing sarcoma (ES) is a bone and soft tissue neoplasia, mainly occurring in children and young adults. Molecularly, it is characterized by a chromosomal translocation generating a driver fusion gene between the gene EWSR1 (Ewing Sarcoma Breakpoint Region 1) and one gene from the E26 transformation-specific or *E*-twenty-six (ETS) family (FLI1 in most of the cases), with few other genomic alterations. Clinical presentation of the disease still varies from patient to patient. While efforts have been made to improve treatment regimens and outcome, patients with metastasis at diagnosis or relapsed disease still present a dismal prognosis [1]. This fact could be directly related with intertumoral and intratumoral heterogeneity (ITH) amongst patients and the clinical and experimental challenges that are yet to be overcome. Clinically, there is an evident need to unveil and test new agents to establish more effective and safer regimen treatments. The major challenges in terms of drug discovery in the field of ES are related with the lack of appropriate experimental models. On the one hand, despite the joint efforts, researchers have not been able to generate an ES mouse model [2]. On the other hand, while other neoplasm organoids or tumoroids are routinely used in pre-clinical studies, the generation of such structures recapitulating the disease is still far ahead for sarcomas. So far, preclinical models are restricted to the use of cell lines and patient-derived xenografts (PDX) [3,4,5,6]. While the generation of PDX models certainly helped increase the confidence level of drug studies, these models also present limitations, namely regarding possible clonal selection along passages and the lack of the appropriate tumor microenvironment, which, in our point of view, might result in variations regarding the transcriptomic profile.

Additionally, although new breakthrough technologies are currently reshaping the way we, researchers, see the molecular phenotype of ES, the new knowledge needs to be accurately validated to guide clinical decisions that might benefit patients. In this review, we will deepen into how the latest works might contribute to a new era in diagnosis and disease management in ES.

## 2. Improvement in the Diagnosis and Discovery of New Entities Using Next-Generation Techniques

Bone and soft tissue tumors are arguably among the most challenging neoplasms for precision diagnostic. The histological classification of soft tissue tumors has traditionally been performed by morphological analysis and immunohistochemistry (IHC); however, many tumors show nonspecific or overlapping marker expression. Specifically, CD99 is a highly sensitive and useful immunohistochemical marker for ES, usually showing a diffuse, strong, membranous pattern of distribution, but it is expressed within a broad variety of mesenchymal tumors as well [7]. Other nonspecific markers observed in ES are S-100 protein, CD57, neurofilaments, cytokeratin, and desmin. Similar to many mesenchymal tumors, ES is characterized by recurrent gene fusions (GFs) with a major role in oncogenesis, which often are tumor specific. Thus, molecular testing to detect gene fusions is mandatory for the correct pathological diagnosis. Currently, a wide range of GFs and GF variants have been described in ES, although the vast majority of ES harbors the fusion of the EWSR1 gene (a member of the FET family comprising FUS, EWSR1, TAF15 genes which contain an RNA-binding domain) [8] with the FLI1 gene (a member of the ETS transcription factor family) on 11q24 [9]. Alternative fusion transcripts in ES involve EWSR1 or FUS with other members of the ETS family (ERG being the second most common, followed by ETV1, ETV4, and FEV) [10,11,12,13]. EWSR1 and FUS appear to be functionally interchangeable. Occasionally, EWSR1 is fused to non-ETS gene partners (PATZ1, SP3, NFATc2, and SMARCA5) defining a subgroup of tumors with atypical morphology [14,15], which is currently denominated *round cell sarcoma (RCS) with non-ETS fusions* (being *EWSR1*-*NFATC2*, *FUS*-*NFATC2,* and *EWSR1*-*PATZ1* the most common GFs). FLI1 and ERG immunopositivity can be seen in those ES harboring *EWSR1-FLI1* and *EWSR1-ERG* GFs, respectively [16]. Interestingly, the expression of PAX7 has recently been shown to be restricted to those tumors demonstrating a fusion between EWSR1 and FLI1, ERG, and NFATc2 [17]. Some small round cell sarcomas previously considered atypical subtypes of Ewing sarcoma are genetically and clinically distinct entities and include CIC (Capicua Transcriptional Repressor)-rearranged sarcoma and sarcoma with BCOR (BCL6 corepressor) genetic alterations [14,18]. All these entities often exhibit deceptive and overlapping histomorphologic features, but show a different clinical behavior [18,19], so it is crucial to perform an accurate differential diagnosis (Figure 1).

Despite widespread use of molecular testing with traditional gold standard techniques, such as fluorescence in situ hybridization (FISH) and/or reverse transcriptase-PCR (RT-PCR), it can be challenging to attain a precise diagnosis of ES. Break-apart FISH is more widely available, because it requires a small amount of tissue, has a fast turnaround time, and does not require a priori knowledge of the two gene partners. However, *EWSR1* break-apart FISH can be particularly tricky to analyze, as the tissue can be crushed and the signals overextended, and intrachromosomal rearrangements are often undetectable [20,21]. Furthermore, the accurate differential diagnosis of ES may require assessing a sizable variety of GFs with different exonic variants as well, and these methods do not allow the simultaneous evaluation of multiple GFs. Thus, repeated FISH probing has to deal with sample exhaustion, which is a common issue since sampling techniques usually minimize tissue availability.

Over the last two decades, improvements in molecular techniques have provided important general insights and greatly contributed to improving the differential diagnosis of ES and related entities. In this context, next-generation sequencing (NGS)-based approaches are currently being used as an efficient ancillary technique [22,23]. As NGS is based on a multiplex assay, it saves time and minimizes the consumption of tissue material. The targeted-RNA sequencing method based on Anchored Multiplex PCR (AMP) (Archer FusionPlex Sarcoma assay) is commonly used today, preferring RNA to DNA as starting material because most of the GFs arise due to breaks within large introns. Furthermore, the amplification using both universal and gene specific primers elicits GF identification without prior knowledge of fusion partners, contributing to the discovery of novel GFs and/or variants.

NanoString nCounter platform represents another alternative for the multiplexed testing of GFs. The NanoString nCounter assay is a high-throughput hybridization technique using target-specific probes that can be customized to test for many fusion transcripts in a single assay using RNA from formalin-fixed, paraffin-embedded material. Chang KTE et al. have designed a NanoString assay targeting 174 unique fusion junctions in 25 sarcoma types [24]. The study cohort comprised 212 cases, 96 of which showed fusion gene expression by the NanoString assay, including all 20 ES, 11 synovial sarcomas, and 5 myxoid liposarcomas. Among these 96 cases, 15 showed fusion expression not identified by standard clinical assays. There were no false-positive results; nevertheless, four cases were false negative when compared with FISH or RT-PCR. Another NanoString assay for testing 22 fusion transcripts associated with the most prevalent pediatric sarcomas was developed by Javal Sheth et al. [25]. The results showed that NanoString assay was 100% concordant with RT-PCR. A third study using NanoString for the detection of sarcoma GFs has been currently published by Wangzhao Song et al. [26]. A cohort of 104 soft tissue tumors representing 20 different histological types was analyzed for the expression of 174 unique GF transcripts. A tumor-defining GF transcript was detected in 60 cases (58%). The highest diagnostic coverage was obtained for ES, synovial sarcoma, myxoid liposarcoma, alveolar rhabdomyosarcoma, and desmoplastic small round cell tumor. Therefore, this approach shortens the turnaround time and reduces the reagent cost per sample compared with conventional techniques, and could serve as a first-line clinical diagnostic test for sarcoma GF identification, replacing multiple single plex assays. However, when no fusion event is identified, the targeted-RNA sequencing method based could still detect novel GF and/or variants, but with higher costs and longer turnaround times. To conclude, according to the advantages and disadvantages of the diagnostic tools described, we propose an algorithm for differential diagnosis of ES and related entities, which could be followed in the clinic to improve the sensitivity and specificity and reduce the consumption of sample material, cost per sample, and the turnaround time (Figure 2).

## 3. The Heterogeneous Molecular Phenotype of Ewing Sarcoma

Genomic alterations such as the loss of 16q and gain of chromosome 8 in 50% of the cases, 1q gains in 25% of the cases, and the microdeletion of p16 and mutation of TP53 and CDKN2A in around 20% of cases have been extensively described in the literature and are reviewed comprehensively by Grunewald et al. [1,27]. Some of these alterations are currently undergoing prospective validation as prognostic biomarkers in clinical trials. Nonetheless, in the last five years, new lines of thought have opened up when it comes to sarcoma research due to the introduction of cutting-edge sequencing technologies. Not only new types and subtypes of sarcomas bearing GF have been discovered and molecularly characterized, but also, the relevance of secondary alterations and molecular intertumoral and ITH has been addressed [19,28,29,30,31,32,33,34,35]. More recently, using NGS, three studies described the presence of STAG2 (cohesin subunit SA-2 protein) mutations or rearrangements in a subgroup of ES cases. Crompton and colleagues showed that STAG2 mutation, resulting in a loss of expression, was associated with metastatic disease. Indeed, 88% of the patients of STAG2 loss in the primary tumor presented metastatic disease. Apart from STAG2, previously described mutations in TP53 (tumor protein p53) and CDKN2A (cyclin dependent kinase inhibitor 2A) were also found, both in tumors and cell lines, in accordance with what had already been described. Authors describe how by using RNA sequencing, whole genome sequencing (WGS), and single nucleotide polymorphism (SNP) arrays, ES presented few genetic alterations [29]. Another interesting result from this work is the presence of complex genomes in relapse when compared to the primary tumors. Almost simultaneously, Tirode and colleagues performed a similar study [35]. Here, DNA from 112 ES patients and germline DNA were sequenced by WGS. Again, the presence of STAG2 and CDKN2A mutations was detected in a small set of patients (17% and 12%), and these were mutually exclusive. STAG2 loss of expression was detected in three cases when comparing primary to relapsed tumors. In these cases, STAG2 mutations were detected with higher allelic fractions than in the primary tumor at diagnosis.

Finally, in a third work published by Brohl and colleagues, a set of 65 samples from the ES family of tumors and 36 cell lines were sequenced using WGS and targeted-RNA sequencing [28]. Authors found that, once more, a loss of expression of STAG2 by mutation was observed in 21.5% of the cases and in 44.4% of cell lines. Loss of expression was confirmed by IHC and associated with advanced disease. The statistical association with overall survival was lower than in the previous studies, which was probably due to the lower number of samples employed in the study. In conclusion, ES presents a rather quiet and flat genomic profile at the moment of diagnosis. However, this is not observed post-treatment. Post-treatment tumors showed a more complex genomic profile and were significantly different from the initial tumors, suggesting the presence of a clonal selection during treatment. The loss of STAG2 seems to have a critical biological role in the development of metastasis, hence affecting disease progression.

The generation of latent EWSR1-ETS translocations and the presence of clonal evolution in ES with chromoplexy have been recently suggested by Anderson et al. [36]. Chromoplexy is a complex process that is still not fully understood, by which replication-activated areas of the genome might suffer rearrangements with other areas of genome, occasionally giving rise to gene translocations. This process has been previously described in prostate cancer by the presence of *TMPRSS2-ERG* [37]. In this remarkable work, authors showed that ES tumors driven by chromoplexy presented a defined clonal evolution where primary and relapsed disease followed different patterns. Tumors could develop years before detectable disease was present. In addition, these presented a worst prognosis [36].

In terms of epigenetics, Riggi et al. described, by integrate chromatin state analysis, that EWSR1-FLI1 induces de novo enhancers and represses others, demonstrating that epigenetic reprogramming is key in the ES transcriptional program [38]. In 2016, Huertas-Martinez and colleagues showed that by profiling the DNA methylation of 15 samples, normal tissue, Mesenchymal Stem Cells (MSC), and tumor cell lines that PTRF, a gene involved in caveolae formation, was particularly relevant [30]. The authors showed that upon restoring the expression of this gene, ES cells engage in a cascade that culminates in apoptosis. This supports the idea that the epigenetic regulation of ES is critical for its transcriptional program. Later on, a larger study on the DNA methylation pattern of ES showed clear intertumoral heterogeneity, as described by Sheffield and colleagues where ES were clustered by their methylation profile and compared to other tumors and the putative cell of origin, the MSC [34]. This study showed that ES presents a heterogeneous profile with a medium to high range of coefficient variation, which provides a ratio to compare intertumoral heterogeneity. Surprisingly, using robust bioinformatic methods, the authors found that ES had high levels of ITH. Likewise, patients with metastasis at diagnosis presented higher ITH when compared to patients with local disease.

So far, little is known about the presence and the clinical and biological value of this ITH and further studies need to be performed to evaluate how this might affect resistance and contribute to the process of dissemination. Until now, the presence of high levels of the EWSR1-FLI1 had been associated with the presence of malignant phenotype. However, in 2018, Franzetti and colleagues showed that ES cells present different levels of the EWSR1-FLI1 transcript within the same cell line, using single cell sequencing. Once the fusion protein is primarily considered as the major driver of the transcriptional program, the presence of differential levels of the fusion transcript within the same cell line represents a major finding [39]. The authors suggest that the ES cells can modulate the levels of EWS-FLI and that these results in a shift in cell plasticity toward a more invasive phenotype in the presence of low levels of the fusion protein. Given that the presence of metastasis defines a clear group associated with worst prognosis, this represented a breakthrough in the field. 

## 4. Advances and Utility of Liquid Biopsy-Based Studies

Liquid biopsies (LB) have attained enormous relevance in the field of cancer diagnosis and monitoring, particularly in the last few years. The advent of technologies with high sensitivity together with the non-invasive trait of LB has had a real impact in the design and validation of new clinical applications [40]. Highly proliferative tumor cells liberate vesicles, which constitute a snapshot from the tumor and fragments of DNA or RNA due to necrosis or apoptosis. These eventually reach the bloodstream as extracellular vesicles, cell-free or circulating-tumor DNA or RNA (ctDNA or ctRNA), respectively. In addition, some cells (circulating tumor cells, CTC) are known to detach from the primary tumor alone or in clusters and extravasate into the bloodstream, circulate, and eventually home at a distance and cooperate to create a micrometastasis. In the case of ES, recent studies have been fundamentally focused on the potential of detectable ctDNA/ctRNA, or extracellular vesicles to improve diagnosis and facilitate disease management. 

Krumbholz and colleagues showed by studying xenograft models and patient´s samples, the detection of EWSR1 fusions ctDNA and its correlation with tumor burden. In 20 ES patients, ctDNA was evaluated during the treatment by droplet digital PCR (ddPCR) using as a probe the specific GF variant present in the primary tumor. Fluctuations on the ctDNA of EWSR1 fusion were correlated with response to treatment, ctDNA was reduced after chemotherapy, and early relapse was detected by an increment on the ctDNA levels in patients who later developed metastases. This biomarker was more sensitive in the detection of early relapse and minimal residual disease than PET/SCAN, which represents the current standard method [41]. Shukla and colleagues also showed that plasma cell-free DNA represents an important resource to monitor disease [42]. The specific breakpoints in the primary tumors were sequenced by NGS in the DNA of patients with ES ( *n* = 11) and desmoplastic small round blue cell tumors (DSRCT) (*n* = 6) (another t-sarcoma, bearing EWSR1-WT1 translocation), along with the recurrent mutations of relevant genes such as TP53, STAG2, and CDKN2A. Plasma cell-free DNA was tested using two different techniques: ddPCR and custom capture NGS. By ddPCR, fusion detection in ctDNA was successful in all ES patients at baseline, while NGS was less sensitive detecting 10/11 DNA fusions. While NGS was slightly less sensitive, the detection of DNA from the translocation was achieved by deep coverage without prior knowledge of the precise breakpoint. In contrast, identification of the breakpoint in the primary tumor is necessary for the design of ddPCR-targeted probes.

The Children’s Oncology Group also showed that the study of ctDNA burden might be useful in the clinical setting, even in the absence of the primary tumor [43]. Using NGS hybrid capture and ultra-low-pass whole-genome sequencing, ctDNA from 94 ES and 72 osteosarcoma patients from banked plasma samples were analyzed. ctDNA was detected in 52.1% of ES patients (EWSR1–ERG *n* = 5; EWSR1–FLI1 *n* = 43) and could be associated with lower event-free survival and overall survival. ES patients with metastasis at diagnosis with detectable ctDNA presented inferior event-free survival than patients with no detectable ctDNA. Similarly, also here, STAG2 and TP53 mutations (in 3 and 4 patients, respectively) were detected using these methods, along with the presence of a novel translocation (EWSR1–CSMD2). These studies showed that both at a pre-clinical and clinical level, ctDNA, particularly from the translocation, is a useful tool for diagnosis and disease monitoring.

In terms of ctRNA, Allegretti and colleagues showed that the detection of EWSR1–FLI1 mRNA by ddPCR in the primary tumor and plasma represented a potential biomarker for ES monitoring. A robust method using ddPCR detected different EWSR1-FLI1 transcript variants in the 5 primary tumors and in 4 blood samples [44]. Four patients with different courses of treatment (adjuvant chemotherapy and surgery; neoadjuvant chemotherapy and radiotherapy; neoadjuvant chemotherapy, surgery and adjuvant chemotherapy) were properly followed and studied. Herein, blood from two patients was collected at diagnosis and after treatment; one patient’s blood was collected only after at the time of surgery, and another one was collected at diagnosis, prior to and after surgery, and after adjuvant chemotherapy. PET/SCANS were available at diagnosis and before surgery. The presence of the transcript was correlated with the volume of the tumor. This study showed that the fusion ctRNA could be a potential biomarker to follow treatment and clinical response. However, one might argue that the presence of ctRNA is probably lower and less stable than the presence of ctDNA. Altogether, we believe that ctDNA appears to be a reliable tool to follow treatment; nonetheless, studies with a higher number of patients should be performed to validate this biomarker.

Tumor cells secrete extracellular vesicles (EVs)**,** representing the background of the tumor, which act as primary messengers to the surrounding tissue to remodel specific areas to accommodate CTC, and thus contribute to the formation of micrometastasis [45]. This discovery opened a new window in terms of LB clinical applications and has since been the object of a great number of studies, especially in tumors with specific mutations. Although normal cells also release EVs, some traits such as certain single-nucleotide variants (SNVs) are indeed detectable in EVs isolated from plasma. To this end, Grunewald et al. firstly reported the characterization of the transcriptional profile of EVs secreted by ES cell lines and how this might be helpful in the field of LB. Herein, the authors report that although EVs carry the mRNA from the fusion transcript, the fusion protein itself was not detectable [46]. More recently, it has been shown that exosomes derived from ES cells contribute to the immunosuppressive profile and the process of inflammation by being internalized by cells of the immune system [47]. Plasma EV RNA was evaluated by RNA sequencing, and in 12 samples, enrichment in exonic non-coding and intergenic RNAs, which are associated with viral retro elements, were detected.

Overall, we foresee that the inclusion of LB-based technologies in the clinical setting is of great value in sarcoma disease management. The incorporation of these technologies will enable patients to be monitored through non-invasive techniques, hence improving cost-effective clinical decisions regarding follow-up and early relapse.

## 5. Unveiling New Molecular Targets Based on Pre-Clinical Studies

The treatment of ES tumors has been based on conventional chemotherapy, radiation, and surgery. Despite the “good” response of localized tumors, new therapeutic regimens must be implemented for the management of relapses and aggressive metastasis, which are frequently associated with patient’s death. The pathogenesis of ES is mainly related with the EWSR1-FLI1 chimeric gene, which is an aberrant transcription factor that promotes changes in gene expression and malignant transformation [48]. Given the dependency of ES cells on EWSR1-FLI1, this oncogenic fusion gene is a very interesting drug target explored by different strategies [49]:

(i) Splicing inhibitors. The inhibition of spliceosome factors using drugs such as Pladienolide B (SF3b1 inhibitor) affects EWSR1-FLI1 pre-mRNA processing, reducing the expression of upregulated genes as a cause of the translocation [50].

(ii) Minor groove-binding agents. Drugs such as Lurbinectedin reduce the binding of EWSR1-FLI1 to the DNA, redistributing the fusion protein within the nucleus [51]. In 2018, the results of a phase 2 clinical trial showed a potent response of advanced ES tumors to Lurbinectedin in monotherapy [52].

(iii) RNA helicase A and EWR1-FLI1 interplay inhibitors. The interaction of EWSR1-FLI1 with RNA helicase A (RHA) is essential for the transcriptional activity of the fusion protein [53]. Small molecules with the ability to block EWSR1-FLI1 binding to RHA have been developed. Amongst them, YK-4-279 (also known as TK216) was able to reduce ES tumor growth *in vitro* and in PDX models [54]. YK-4-279 has also been tested in a clinical trial (NCT02657005) for patients with relapsed or refractory ES.

Despite the diversity of studies harnessing EWSR1-FLI1 as a druggable target, our knowledge about how tumor cells respond to the decrease of EWSR1-FLI1 activity is limited. Franzetti and collaborators recently published a study describing that the low expression of GF was associated with the loss of adhesion proteins and increased cell migration [39]. The phenotypic plasticity of ES tumor cells upon changes in the GF level needs further confirmation, as it may have an important impact in the clinical management. Some of the previous drugs targeting EWS-FLI1 activity could be contraindicated for ES treatment, since they could induce the metastatic process. During the last years, ES tumor biology has been extensively studied using different novel approaches in order to discover new druggable pathways. Massive drug tests, global tumor mutations and epigenetics analysis, single-cell multiple omics, and CRISPR-Cas9-based screenings have emerged as important tools to identify therapeutic targets. In some cases, advances in tumor cell biology knowledge and its vulnerabilities have enabled the development of clinical trials with promising results. The most interesting targets, strategies for discovering, and potential therapeutic regimens are described below.

PARP1. Poly (ADP-ribose) polymerase 1 (PARP1) plays a crucial role in the repair of different kinds of DNA damage including single and double-strand breaks (SSBs, DSBs). The detection of lesions by PARP1 triggers a fast-cellular response involving the recruitment of several repair factors [55]. The relevance of PARP1 in the accurate repair of DNA damage has motivated the examination of PARP inhibitors as antitumoral agents. Recent studies have suggested that PARP inhibition induces the accumulation of DSBs, which is the most cytotoxic form of DNA damage [56]. As a consequence, tumors with mutations in DSBs repair factors such as BRCA1/2, which are members of the homologous recombination (HR) pathway, are especially sensitive to PARP inhibitors (PARPi) [57]. The hypersensitivity of ES cells to PARPi was described by Garnett and colleagues as a result of a high-throughput screening [58]. This effect is also related to an HR deficiency in the absence of BRCA mutations. Gorthi and colleagues described a “BRCAness” phenotype mechanism in which the high transcription rate of ES cell lines is associated with R-loops (RNA:DNA hybrids generated during transcription process) accumulation and BRCA1 recruitment. The role of BRCA1 in R-loops resolution is associated with a lower efficiency of the HR pathway [59]. Despite PARPi-mediated lethality in ES cell lines, a not significant response was observed in patients treated with classical inhibitor Olaparib [60]. The discovery and validation of new predictive biomarkers of PARPi sensitivity or resistance could help us to better select patients for future clinical trials with combinatory regimens where PARPi (olaparib or others) could play an important part. Nowadays, combination with radiation or chemotherapy (irinotecan, temozolomide) seems to be a potent therapeutic strategy explored in several clinical trials [61,62].

DNA replication fork proteins. ES cell lines show a high endogenous level of replication stress due to the accumulation of unresolved R-loops, which blocks replication fork progression [59]. Proteins such as ATR or STAG2, with functions in replication stress response, have been analyzed as ES potential targets. ATR is crucial for safeguarding the genome stability by preventing the breakage of the stalled replication fork [63]. Inhibitors of ATR or its targets (CHK1, MAPKAP-K2) have been extensively explored in clinical trials for many tumors [64]. In ES, ATR inhibitors have been tested as single agents showing high efficacy in mice models [65]. As indicated above, mutations in STAG2 are detected in 15%–20% of ES patients and have also been associated with poor prognosis [35]. STAG2 interacts with replication intermediates, and its absence induces replication fork collapse and DNA damage accumulation [66,67]. STAG2 loss-of-function mutations sensitize cancer cells to inhibitors of DNA repair factors (ATR, PARP1) and increase sensitivity to select cytotoxic chemotherapeutic agents [33].

CDKs. Cyclin-dependent kinases (CDKs) are serine/threonine kinases with important roles in the control of cell cycle and transcription [68]. The relationship between CDKs and cancer has been largely explored. Mutations in CDKs and their regulators are related with uncontrolled proliferation and chromosomal instability [69]. An example of how novel methodologies can help to determine ES dependency pathways is the study published by Kennedy and collaborators. The profiling of super-enhancer by chip-seq, in combination with shRNA and drug screenings, revealed that cyclin D1/CDK4 could be a potential therapeutic target [70]. Recently, the same group found, using a gain-of-function screening with open reading frames (ORF), that IGFR (Insulin Like Growth Factor 1 Receptor) overexpression is involved in CDK4/6 inhibitors resistance [71]. These approaches enable us to discover drug combinations in order to bypass chemotherapy resistance mechanisms. In addition, high-throughput screening in ES cell lines has also determined that the expression of EWSR1-FLI1 confers sensitivity to CDK7/12/13 inhibitors. Combination with olaparib has a remarkable effect, reducing tumor growth and increasing survival in PDX mouse models [72].

Regulators of p53 activity. As it has previously been indicated, ES is characterized by a very low mutation incidence. The wild-type TP53 gene is expressed in 90% of tumors [73]. Consequently, the p53 pathway has emerged as a powerful therapeutic target with benefits for the majority of patients. CRISPR-Cas9 screen performed by Stolte and collaborators in ES TP53 wild-type cell lines identified druggable candidates with roles in p53 regulation. The inhibition of MDM2 (involved in p53 degradation) or MDM4 (a p53 inhibitor) enhances p53 activation, reducing tumor growth in “in vivo” models [74]. In our opinion, the combination of these inhibitors with p53 activators such as genotoxic agents or radiation could be potent therapeutic approaches in ES treatment.

GGAA microsatellites-associated genes. A microsatellite is an in-tandem repeat of a short sequence of nucleotides (1–6 pb) with a variable number of repetitions [75]. The transcriptional regulation of some EWSR1-FLI1 target genes has been related with the binding of the fusion protein to upstream GGAA microsatellites [76]. Recently, Musa and collaborators described the importance of length polymorphisms of these GGAA microsatellites for tumor growth, survival, and drug response. A clinically relevant candidate shown in this study is MYLB2 (also known as B-MYB), which has been implicated in cell cycle regulation, cell survival, and the differentiation/maintenance of stem cell phenotype [77]. The inhibition of CDK2, an upstream-activating kinase of MYLB2, reduces tumor growth in an ES mice model [78].

Immunotherapy. The development of cancer immunotherapy has been associated with the discovery of specific tumor antigens. Nowadays, immunotherapy is focused on different approaches, including the development of monoclonal antibodies, adoptive cell therapy, cancer vaccines, and cytokines [79]. In the case of ES tumors, our group published recently a pre-clinical study using a drug–conjugate antibody against the transmembrane protein endoglin (ENG). In this work, we linked nigrin-b (an inactivating-ribosome protein) and cytolisin (a pore-forming protein) to the antibody, showing encouraging results both in vivo and in in vitro [80]. From our point of view, the combination of anti-ENG antibodies with other drugs, such as genotoxic agents, could be interesting for increasing treatment specificity and reducing toxicities.

Another approach is the generation of TCR transgenic T cell receptor for the recognition of chondromodulin 1 (CHM1), which is an ES overexpressed protein involved in invasion and metastasis [81]. Finally, programmed cell death receptor 1 (PD-1) and its associated ligand (PD-L1) have also been checked as immunotherapy targets. The role of these proteins in T cell inactivation has motivated the use of anti-PD1 antibodies such as nivolumab or pembrolizumab in order to enhance the antitumoral activity of the immune system [82]. Different clinical trials are currently ongoing for ES patients, amongst other sarcomas, using these antibodies. Of note, ES has often been characterized as a ‘cold tumor’ with low immunogenicity. In fact, given that ES presents a very low mutation rate/tumor mutation burden, the possibility of generating CAR-T recognizing neo-antigens might be limited.

## 6. Future Directions on Ewing Sarcoma Research

Significant therapeutic advances arrived at the clinical arena four decades ago and benefited some patients with ES. However, ES is a clinically heterogeneous neoplasm. In fact, there are fundamentally two groups of patients within ES: (i) those with multifocal and disseminated involvement, representing approximately 40%, and (ii) those with localized disease. For the last 25 years, research has been focused on understanding the biological background that leads to this clinical heterogeneity [83]. However, we have very few answers at the moment, and perhaps that is the first frontier of knowledge in ES research. In this review, we have provided a discussion on some technological advances that will surely generate a wealth of biological and clinical information. An accurate diagnosis through diagnostic platforms such as the one described in this review, together with enhanced follow-up methods (LB), is the basis of patient selection for innovative therapies. The latter require the application of new and more accurate experimental models that would allow systematic and controlled study of the impact of drugs at the level of the individual cell, and more particularly the patterns of sensitivity and resistance to genotoxic agents, the involvement of the stroma and immune system on drug response, and the possible phenotype of the cells with self-renewal capacity. Another relevant aspect in the use and generation of experimental models is the fact that ES might arise in different type of tissues (bone and soft tissue). As the site of origin of this tumoral entity varies, so does the tumoral microenvironment, creating thus another variable that must be considered when evaluating drug administration, release, uptake by the tissue, and secretion.

Our hope is that this technical knowledge can generate a wave of new therapeutic possibilities. The gap between the richness of biological knowledge and the lack of new treatments is striking. Perhaps that is why the other big challenge in ES research is the frontier of clinical research. That is, the new therapeutic protocols should select drugs with a clearer rationale to break the biology of ES and incorporate biomarkers that have been previously validated in retrospective studies. An obvious proposal is that all future patients with ES can benefit from the determination of STAG2, 1q gains, and 16q losses, among others [27,35]. Almost mandatory to all clinicians who diagnose and treat ES is the collection and storage of biological samples to develop the research of the future, and the precision diagnosis that our patients require today. The incorporation of next-generation techniques together with the availability of well-collected samples and data, to the day-to-day research will allow investigators to move further and further into a better understanding on the biology of this devastating disease.

## Figures and Tables

**Figure 1 cells-09-00804-f001:**
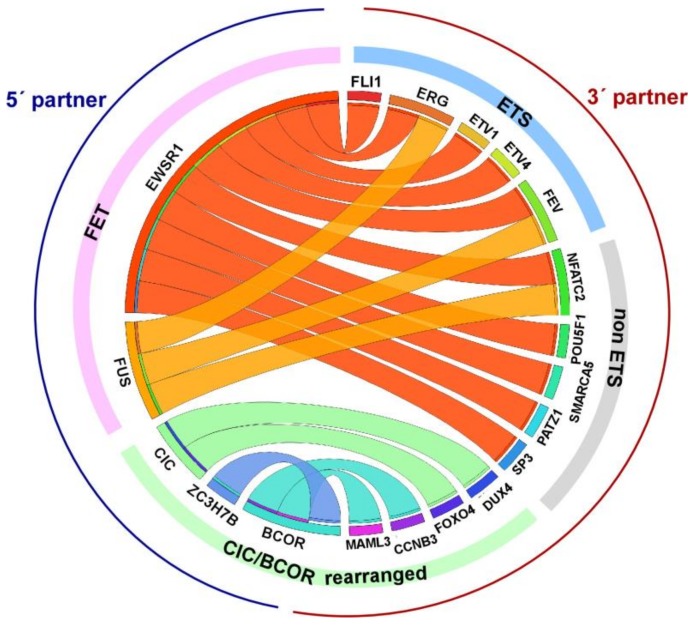
Circos plot depicting GFs in Ewing Sarcoma (ES) and related entities. Canonical ES GFs comprise fusions between members of the FET family of RNA-binding proteins (EWSR1 and FUS) and the ETS family of transcription factors. *Non-ETS* translocated genes as well as CIC and BCOR GFs are also indicated. BCOR internal tandem repeats are not represented. The 5′ partners are indicated in blue, whereas the 3′ partners are highlighted in red.

**Figure 2 cells-09-00804-f002:**
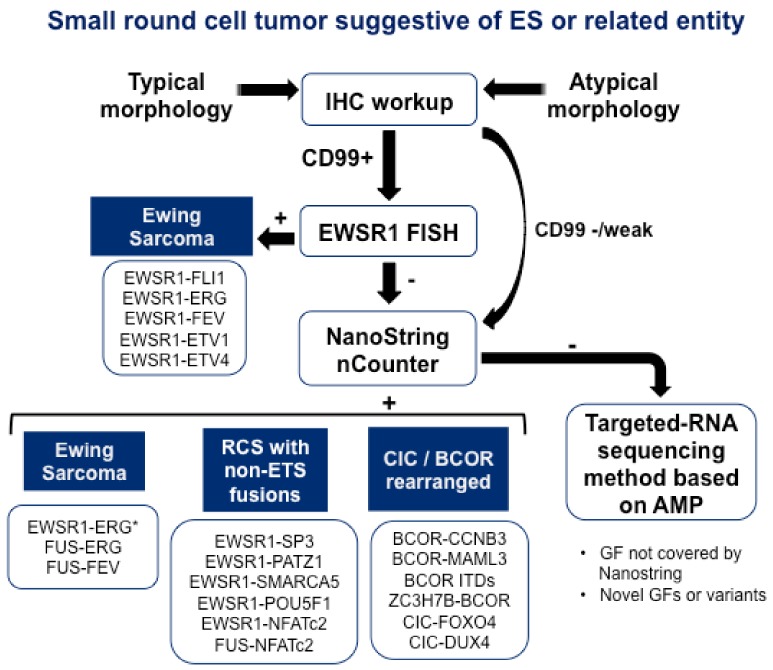
Proposal of an algorithm for the differential diagnosis of ES and related entities. The morphologic analysis by hematoxylin and eosin staining and immunohistochemistry (IHC) workup are the first line diagnostic tests. Canonical ES cells are characterized by CD99 expression by IHC and a uniform and small round appearance with round nuclei, while atypical ES cells show negative or weak CD99 staining and are larger, with prominent nucleoli and irregular contours. Our proposed algorithm consists of the following steps. First, IHC workup should be performed to rule out sarcomas with EWR1-rearrangement that can show a similar round cell morphology (i.e., high-grade myxoid liposarcoma, myoepithelial carcinoma). An EWSR1 fluorescence in situ hybridization (FISH) test is performed in CD99 positive samples, and if EWSR1 rearrangement is positive, canonical ES can be diagnosed. However, when CD99 IHC is negative or weak and/or the EWSR1 FISH test does not provide clear results, NanoString methodology is presented as the next pertinent diagnostic tool. This multiplex technology allows the detection of ES harboring GFs that involves FUS and non-FET genes. NanoString assay is also convenient to detect ES harboring EWSR1–ERG* fusion, because FISH assays may not be able to detect this gene rearrangement due to the complex pattern of t [21,22] translocation [20]. Moreover, CIC-rearranged sarcoma and sarcoma with BCOR genetic alterations can be identified. Finally, if NanoString renders negative or non-evaluable results, samples should be tested by targeted RNA sequencing based on AMP to detect GFs not covered by the NanoString assay.

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
