# Peer review of "Breakthrough Technologies Reshape the Ewing Sarcoma Molecular Landscape"

_cells, 2020, doi:10.3390/cells9040804_

Round 1

Reviewer 1 Report

This review aims to discuss how the latest progress of new technologies could contribute to understanding the mechanism and provide a potential treatment for Ewing’s Sarcoma. Overall, this review summarized the current research advance and provided suggestions in ES research and treatment. I have a few comments, as listed.

“The major challenges in terms of drug discovery in the field of ES are related to the lack of appropriate experimental models.” Please explain why the PDX models are not sufficient for ES drug discovery and how the new technologies could help with better animal models?

Ewing’s sarcoma is a highly malignant bone tumor. I recommend the author discuss the tumor precursor cells and the bone homeostasis in this review.

In ES, fusion-derived NeoAgs (e.g., EWSR1-FLI1) may be a target for immunotherapy. I highly recommend the author briefly summarize current progress and discuss the NGS technology and the potential contribution to ES immunotherapy.

Abbreviations, please reference the full name first in the body of the text with the abbreviation in parenthesis. Please check all the Abbreviations, and one recommendation site is https://www.jci.org/kiosks/publish/abbreviations. Please only use the standard abbreviations. For example, GF is very confused for me, Gene fusions? Growth factor?

Author Response

Our responses are in italics:

This review aims to discuss how the latest progress of new technologies could contribute to understanding the mechanism and provide a potential treatment for Ewing’s Sarcoma. Overall, this review summarized the current research advance and provided suggestions in ES research and treatment. I have a few comments, as listed.  We thank the reviewer for his/her comments

“The major challenges in terms of drug discovery in the field of ES are related to the lack of appropriate experimental models.” Please explain why the PDX models are not sufficient for ES drug discovery and how the new technologies could help with better animal models?

An explanation for the limitations of PDX models has been included in the Introduction (page 2, rows 44-47)

Ewing’s sarcoma is a highly malignant bone tumor. I recommend the author discuss the tumor precursor cells and the bone homeostasis in this review.

We think this extremely interesting issue is out of the scope of the manuscript, although lines 206-227 deal in part with the mechanisms of sarcomagenesis in Ewing sarcoma

In ES, fusion-derived NeoAgs (e.g., EWSR1-FLI1) may be a target for immunotherapy. I highly recommend the author briefly summarize current progress and discuss the NGS technology and the potential contribution to ES immunotherapy.

In lines 426-429 we have made a brief comment on this aspect

Abbreviations, please reference the full name first in the body of the text with the abbreviation in parenthesis. Please check all the Abbreviations, and one recommendation site is https://www.jci.org/kiosks/publish/abbreviations. Please only use the standard abbreviations. For example, GF is very confused for me, Gene fusions? Growth factor?

We have checked abbreviations, included GF (growth factor) and avoided them when possible

  Thank you and best wishes

Reviewer 2 Report

The paper is very well written and very clearly highlights important molecular news in Ewing sarcoma. However, I suggest the authors to better clarify the following points.

1) In the text and figures the authors should use the current names for "non-Ewing" sarcomas: "Atypical Ewing sarcoma" shoud be replaced by "Round cell sarcoma with EWSR1-non-ETS fusions" (maybe using an acronym), and "Ewing-like sarcoma" should be replaced by "CIC-rearranged sarcomas / sarcomas with BCOR genetic alterations". Thus due to the fact that "Atypical Ewing sarcomas" and "Ewing-like sarcoma" are obsolete; moreover "Atypical Ewing sarcomas" can be referred to "real" Ewing sarcoma (genetically confirmed) with atypical morphology (i.e. with epithelial morphology).

2) In the group of "Round cell sarcoma with EWSR1-non-ETS fusions", the most frequent entities are Sarcoma with EWSR1-NFATC2 fusions; sarcoma with FUS-NFATC2 fusions; sarcoma with EWSR1-PATZ1 fusions. The author shoud better clarify this in the text and in Figure 1 legend.

3) The aurhors can consider the possibility to expand the algorithm in Figure 2 and the corresponding text by introducing RT-PC that can detect a complete gene fusion (i.e. EWS-FLI1 and not only EWR-rearragment) and is cheaper than other NGS tests as nanostring or RNAseq.

4) In Figure 2 and in the corresponding text, it is suggested that a case with EWSR1-rearrangement and strong CD99 positivity can be diagnosed as Ewing sarcoma.

Considereing that EWSR1 is shared by other round cell sarcomas and CD99 is not completely specific, the diagnosis of Ewing sarcoma - with only EWS and CD99 - at least in soft tissues, should be made after ruling out, with immunohistochemistry and genetic, other sarcomas with EWR1-rearrangement that can show a similar round cell morphology (i.e. high grade myxoid liposarcoma, myoepithelial carcinoma...). The authours can add this at least in the Figure 2 legend.

Author Response

We thank the reviewer for her/his comments. A new version, with tracked changes addresing all comments, is provided.

Sincerely,

Enrique de Álava